# Influenza-associated hospitalisation, vaccine uptake and socioeconomic deprivation in an English city region: an ecological study

Daniel Hungerford,[1,2] Ana Ibarz-Pavon,[1] Paul Cleary,[2] Neil French[1]

[1]The Centre for Global Vaccine Research, Institute of Infection and Global Health, University of Liverpool, Liverpool, UK
[2]Field Epidemiology Service, National Infection Service, Public Health England, Liverpool, UK

**Correspondence to**
Dr Daniel Hungerford;
d.hungerford@liverpool.ac.uk

## ABSTRACT

**Objectives** Every year, influenza poses a significant burden on the National Health Service in England. Influenza vaccination is an effective measure to prevent severe disease, hence, maximising vaccine coverage in the most vulnerable is a priority. We aimed to identify the extent to which socioeconomic status is associated with influenza-associated illness (IAI) and influenza vaccine coverage.

**Design** Retrospective observational study using hospital episode statistics.

**Setting** Merseyside, North-West of England, including the city of Liverpool.

**Participants** Residents of Merseyside hospitalised with IAI between April 2004 and March 2016, and Merseyside general practice registered patients eligible for influenza vaccination in 2014/2015 and 2015/2016 influenza seasons.

**Exposures** Socioeconomic deprivation based on lower super output area English Indices of Deprivation scores.

**Primary and secondary outcome measures** Incidence and risk of IAI hospitalisation, and vaccine uptake.

**Results** There were 89 058 hospitalisations related to IAI among Merseyside residents (mean yearly rate=4.9 per 1000 population). Hospitalisations for IAI were more frequent in the most socioeconomically deprived areas compared with the least deprived in adults aged 15–39 years (incidence rate ratio (IRR) 2.08; 95% CI 1.76 to 2.45; p<0.001), 60–64 years (IRR 2.65; 95% CI 2.35 to 2.99; p<0.001) and 65+ years (IRR 1.90; 95% CI 1.73 to 2.10; p<0.001), whereas rates in children were more homogeneous across deprivation strata. Vaccine uptake was lower than the nationally set targets in most neighbourhoods. The odds of vaccine uptake were 30% lower (OR 0.70; 95% CI 0.66 to 0.74; p<0.001) and 10% lower (OR 0.90; 95% CI 0.88 to 0.92; p<0.001) in the most socioeconomically deprived quintile compared with the least deprived, among children aged 24–59 months and 65+ years, respectively.

**Conclusions** Higher rates of IAI hospitalisations and lower vaccine uptake in the most socioeconomically deprived populations suggest that health promotion policies and interventions that target these populations should be a priority.

## Strengths and limitations of this study

► The study uses routine public health data sources and a simple repeatable methodology, which can be applied to continue to monitor changes in incidence and vaccine uptake in local areas, and potentially evaluate targeted population-level interventions.

► We were unable to evaluate population-level impact of influenza vaccines because of non-specific end points and seasonal disease variation over the study period.

► While an association with socioeconomic deprivation was demonstrated for incidence of hospitalisation and uptake independently, difficulties linking data recorded through several incompatible databases prevented integration and development of more complex models.

► Further research is needed to understand the attitudes and beliefs towards influenza disease and influenza vaccination as this will influence vaccine acceptability and uptake among the population.

rates among vulnerable populations worldwide. Although influenza imposes a considerable burden on the National Health Service (NHS) in the UK, estimating the true burden of disease is difficult due to the non-specific symptomatic presentation of disease, and lack of laboratory confirmation. Using routinely collected healthcare system and mortality data, influenza was estimated to account for almost 900 000 general practitioner (GP) consultations, 25 000 hospitalisations, and 20 000 deaths in England and Wales every year.[1] However, several studies attempting to estimate the true burden of disease using different approaches and statistical modelling suggest that burden of illness attributable to influenza in the UK is likely to be underestimated by current surveillance systems.[1–5] Children <5 years of age, those aged 65+ years, and individuals with underlying comorbidities are known to be at higher risk of complicated influenza.[2 6] Socioeconomic

## BACKGROUND

Influenza is a highly infectious disease associated with high morbidity and mortality

deprivation has also been associated with higher influenza-related mortality and hospitalisations.[7–9]

Vaccination is the most effective control measure against complications of seasonal influenza. In the UK, the vaccine is offered free of charge through the NHS to those aged 65 years and over, and to individuals at risk of complicated influenza under 65 years of age. In 2013, the Department of Health (DH) expanded free vaccination to children with an age-phased rollout of an intranasal live attenuated influenza vaccine (LAIV) programme, that at the time of writing, includes all children aged 2–8 years, and will eventually target all those up to 17 years of age.[10] Children aged 2–4 years receive the vaccine at their general practice while those aged 5+ years are vaccinated at school. During the 2016/2017 influenza season, vaccine uptake in England was 70.5% among those aged 65+ years, 36.9% among children aged 2–4 years, 48.6% in risk groups aged 6 months to 64 years, and 55.4% among children aged 5–8 years in school years 1–3.[11 12] However, there is variation in uptake between GP practices and few practices achieve the 40% LAIV uptake target in children aged 2–4 years.[13] There is an inverse association between deprivation and uptake of routine childhood vaccines (eg, vaccines for measles, mumps and rubella, human papillomavirus, influenza).[14–16] Therefore, optimising vaccine delivery and identifying key target areas is necessary to ensure the protection of those most vulnerable, and to maximise cost-effectiveness. Simulation studies supporting the decision to include LAIV in the vaccination schedule indicated that increasing vaccine uptake in children from 30% to 60% can markedly increase both direct and indirect protective effects of the vaccine.[1 17 18] However, these studies assume uniform vaccine coverage and fail to account for the substantial variation in disease risk and vaccine uptake observed across local areas with different socioeconomic status.[15] Given that comorbidities that increase the risk of adverse outcomes from influenza are substantially more prevalent among the most deprived populations,[19 20] it would be expected that ensuring vaccine delivery to the most vulnerable groups would increase cost-effectiveness. Understanding how vaccine uptake varies across local areas with differing socioeconomic status, and correlating this to disease burden and use of healthcare resources, will provide evidence to inform immunisation policy and assist in developing targeted vaccine interventions to mitigate this effect. The aim of this study was to describe the burden of severe influenza-associated illness (IAI) in secondary care across local communities in a defined population in relation to small area level socioeconomic deprivation and vaccine uptake levels.

## METHODS
### Population and setting
The study population was the metropolitan area of Merseyside, North-West England, defined in this study as the local authority areas of Halton, Knowsley, Liverpool, Sefton, St. Helens and Wirral. Merseyside has an estimated resident population of 1.5 million,[21] and is a diverse area that contains some of the most socioeconomically deprived neighbourhoods in England as well as some of the most affluent.[22]

### Data sources
#### Influenza vaccine uptake
Influenza vaccine uptake data for seasons 2014/2015 and 2015/2016 were obtained from ImmForm, the system used by the DH, the NHS and Public Health England (PHE) to record GP vaccination uptake.[21] In addition to vaccine delivery through GPs, since 2015/2016 pharmacies in England have also been able to offer influenza vaccination.[23] To capture these data, pharmacies are required to report any vaccine administration to the patients registered GP, and the GP will input this into the ImmForm system. We extracted aggregated information on vaccine uptake for all groups eligible for vaccination from 288 GP practices in Merseyside. Data included the number of eligible patients and the number of people receiving the vaccine for each eligible group.

#### Hospitalisations for influenza-associated illness
Hospitalisation data were extracted from the national Hospital Episode Statistics (HES) database via PHE, a records-based system maintained by NHS digital, which contains information on all hospital admissions, outpatient appointments and A&E attendances covering all NHS trusts in England.

We extracted anonymised aggregated records of all patients residing in Merseyside admitted as an emergency to a hospital in England with a primary or subsequent diagnosis of acute respiratory illness identified between April 2004 and March 2016. We used the International Statistical Classification of Diseases and Related Health Problems (ICD-10) codes (J0*, J1*, J2*, J3*) to identify acute respiratory infection.[2] We further used the ICD-10 coding to identify confirmed influenza, IAI and comorbidities. Confirmed influenza was identified by codes J09 and J10. IAI codes were selected using the following procedure and criteria:

► Literature search of peer-reviewed publications.[2 24 25]
► Identification of ICD-10 codes used by the NHS trusts as a primary diagnostic of respiratory diseases during the peak influenza season (December to March). We calculated means of daily usage during the peak influenza compared with the rest of year and excluded codes, which had a daily usage ratio of <3 during the peak influenza season compared with the rest of the year.[26]
► Excluding specific ICD-10 codes which were identifying non-influenza-related illness, for example, J211-respiratory syncytial virus (RSV) pneumonia, J123-human metapneumovirus pneumonia and J301-allergic rhinitis due to pollen.
► Excluding ICD-10 codes (J218 and J219) commonly used for RSV. We identified these codes through a

prior study at a major acute care hospital in Merseyside by cross-checking patients with laboratory-confirmed RSV with their ICD-10 primary diagnosis code on discharge (Heinsbroek E, The impact of rotavirus vaccination on hospital pressures in a large paediatric hospital in the United Kingdom, ClinicalTrials.gov NCT03271593).

Final selected ICD-10 codes for IAI were as follows: J06.9 (unspecified acute upper respiratory tract infection), J09 (influenza due to certain identified influenza virus), J10 (influenza due to other identified influenza virus), J11 (influenza, virus not identified), J12.8 (other viral pneumonia), J12.9 (viral pneumonia, unspecified), J22 (unspecified acute lower respiratory tract infection). ICD-10 codes used to identify comorbidities are listed in online supplementary S1-table 1, and were identified using the PHE Flu Plan 2016/2017 and Cromer et al.[2 27]

### Area of residence, population denominators and socioeconomic deprivation

The HES dataset includes a code for neighbourhood area of residence, known as Lower Super Output Areas (LSOAs). There are 32 482 LSOAs in England, each containing approximately 1500 persons.[21] These statistical boundaries were constructed after the 2011 census, and have an associated measure of socioeconomic deprivation: the English Indices of Deprivation 2015, of which the Index of Multiple Deprivation (IMD) is the most commonly used. The IMD is a composite measure of social deprivation based on seven weighted domains: income; employment; health and disability; education skills and training; barriers to housing and other services; crime and living environment.[22] Population denominators at LSOA and Middle Super Output Area (MSOA) were accessed for 2004–2015 by single year of age and sex and aggregated into the appropriate age categories referred to below. This information is published by the Office for National Statistics (ONS) and held by PHE.[21]

### Statistical analysis

Hospitalisation data from HES were recoded, reshaped and aggregated by age, sex, and time before merging to other data sources. Age categories were constructed based on age at hospitalisation: <24 months; 24–59 months; 5–14 years; 15–39 years; 40–64 years; 65+ years.

To analyse the spatial and temporal changes in IAI hospitalisations, we constructed age-adjusted and sex-adjusted standardised incidence ratios (SIRs) for Merseyside at the MSOA level (approximately 5000–12 000 resident population) to avoid small numbers. SIRs that represent the relative risk of IAI hospitalisation in an MSOA were compared with the Merseyside average. To calculate SIRs, we first calculated the expected number of hospitalisations in each $MSOA_{ijk}$ by calculating overall Merseyside rates for each combination of age j and sex k and multiplying these by corresponding MSOA population denominators, as exampled in the following equation:

$$Expected\ number_{ijk} = \frac{MSOA\ population_{jk}}{Total\ population_{jk}} \times Total\ number\ of\ hospitalisations_{jk}$$

Expected numbers are summed for each MSOA and SIRs are calculated by the ratio of the observed to expected numbers in each MSOA, as shown below:

$$SIR_i = \frac{Observed\ number\ of\ hospitalisations_i}{Expected\ number\ of\ hospitalisations_i}$$

We then produced choropleth maps of MSOA SIR for four 3-year pooled time periods for the area of Merseyside.

For analyses focusing on associations between socioeconomic deprivation and hospitalisation or uptake, we first constructed socioeconomic deprivation groups for IMD using quintiles of their score for LSOAs in England. In 2015, 45% of the population in Merseyside lived in the most deprived quintile.[22] Association between socioeconomic status and IAI hospitalisations were assessed by fitting a negative binomial regression model (due to overdispersion) offset by resident population, adjusting for age group, sex and year-by-year variability in influenza incidence by including fiscal year. To examine the association between age group and deprivation, an interaction term was also included. We also stratified by age group and incidence rate ratios (IRR) were calculated along with 95% CIs.

To estimate the association between socioeconomic deprivation and influenza vaccine uptake, uptake data were extracted at GP practice level. We used the LSOA level GP registered population, weighting vaccine uptake by proportion of population served by each GP practice, to create synthetic estimates of vaccine uptake at LSOA level for the 2014/2015 and 2015/2016 seasons. Vaccine uptake levels were classified into three categories in accordance with the national targets set by NHS England, DH and PHE[27]:

► Aged 24–59 months (target: 40%);
► Aged 65+ years (target: 75%);
► At-risk groups under 65 years of age (target: 55%).

The association between socioeconomic deprivation and vaccine uptake was assessed using fractional logit regression and robust SEs.[28] The dependent variable was vaccine uptake, included as a proportion, and the independent variable was IMD quintile. The model was adjusted for fiscal year (April to March) and, because the unit of analysis was LSOA, the model was weighted using LSOA population size. Three separate models were constructed for vaccine uptake in those aged 24–59 months, 65+ years and at-risk groups under 65 years. The models enabled the calculation of ORs and robust 95% CIs.

Demographic characteristics were compared between those with and without clinical risk factors for severe IAI. Continuous variables were tested by Student's t-test, or Wilcoxon rank-sum test if not normally distributed, and $\chi^2$-test or Fisher's exact test were used for categorical variables.

All data processing and statistical analyses were conducted using R V.3.2.3 or above.[29]

## Patient and public involvement

Through our institute's public involvement panel (PIP), the study team has had an ongoing dialogue on influenza vaccine research. This research project was generated from these discussions as PIP members identified that reducing inequalities in influenza vaccine uptake should be a policy priority. The panel have considered and commented on the aims and objectives, and contributed to the development of this study. The findings from this study have been shared with organisations involved in data collection, patient contact and the PPI panel. Investigators have and will continue to present these findings at regional and national events and to participating NHS organisations (including lay membership), public health departments and government agencies.

## RESULTS

### Hospitalisations in Merseyside

Between April 2004 and March 2016, there were 89 058 hospitalisation events coded as influenza or IAI in Merseyside (mean per year: 7421.5; 4.9 per 1000 population), of which 43 767 (49.1%) occurred among male patients (see online supplementary S1-table 2). The highest number of hospitalisations occurred in the 2006/2007 season (n=9247; 6.2 per 1000 population), and the lowest in 2011/2012 (5231; 3.5 per 1000 population). Hospitalisations showed a clear seasonal pattern, peaking in December and January (figure 1). Most hospitalisations (n=29 887; 33.5%) corresponded to patients aged 65 years and over, or children under 24 months of age (n=21 420; 24%). Over the study period, the mean population rate of hospitalisation was highest in children aged under 24 months of age (51.3 per 1000) and 24–59 months of age (15.4 per 1000).

During the 2009/2010 influenza season, when the H1N1 pandemic occurred, the proportion of hospitalisations corresponding to patients aged 65+ years was at its lowest (figure 1; online supplementary S1-table 2), reflecting the shift towards younger adults observed during the H1N1 pandemic. Forty-eight per cent (n=42 712) of hospitalisation had an ICD-10 clinical code that indicated a comorbidity. The two most common comorbidities were heart disease (n=19 293; 45%) and chronic respiratory conditions (n=15 755; 37%). The proportion of hospitalisations with comorbidities for complicated influenza increased with age, and was slightly greater in males (n=22 444; 53%).

### Association between hospitalisation rates and socioeconomic deprivation

Over half of the hospitalisations (n=47 008; 53%) occurred among those living in the most socioeconomically deprived quintile (table 1). Overall, the proportion of hospitalisations for IAI with a comorbidity was lower in the most deprived quintile (47%: n=22 167/47 008; p=0.01) compared with the least deprived (49%: n=2672/5454) (table 1). Regarding age group, a similar proportion of children aged <24 months (range: 6%–7%) and adults aged 65+ years (range 79%–81%) had comorbidities across the deprivation strata (table 1). However, among children aged 5–14 years and among those aged 15–39 years and 40–64 years, the proportion of IAI hospitalisations that had a comorbidity was highest in the most deprived quintiles (table 1). Below the age of 5 years there was little difference in hospitalisation rates between the deprivation quintiles (figure 2), for example, the mean rate of IAI hospitalisations over the study period was 26.5 per 1000 population in the most deprived quintile and 31.4 per 1000 population in the least deprived. For children aged 5–14 years, the risk of IAI hospitalisation was 1.2 times among those from the most deprived compared with the least deprived (IRR 1.23; 95% CI 1.04 to 1.47; p=0.019). In those aged 15–39 years, the risk of hospitalisation was over two times higher (IRR 2.08; 95% CI 1.76 to 2.45; p<0.001) among residents in the most deprived areas in comparison to those in the least deprived quintile; in adults aged 40–64 years, it was 2.6 times higher (IRR 2.65; 95% CI 2.35 to 2.99; p<0.001) and 1.9 times higher in those aged 65+ years (IRR 1.90; 95% CI 1.73 to 2.10; p<0.001). When including an interaction term for the association between IMD quintile and age group, socioeconomic deprivation was strongly associated with an increased rate of hospitalisation in the 15–39, 40–64 and 65+ years age group. Comparing the least deprived IMD quintile with the most deprived and those aged 40–64 years with those aged <24 months, risk of hospitalisation was nearly 2.4 times higher (IRR 2.39; 95% CI 1.94 to 2.95; p<0.001) (see online supplementary S1-table 3). Spatially age-adjusted and sex-adjusted SIRs for IAI hospitalisations also show a visual association with deprivation over time, with higher rates being consistently observed in the most deprived areas (figure 3).

### Influenza vaccine uptake

Vaccine uptake among children aged 2–4 years for the 2014/2015 and 2015/2016 season did not attain the DH target of 40% in most of Merseyside (figure 4A). In 2015/2016, 916/989 (93%) LSOAs did not meet the 40% target. Vaccine uptake among those aged 65+ years in the 2015/2016 season was below the 75% DH target, with 701/989 (71%) LSOA areas below the target (figure 4B). Among those presenting risk factors for complicated influenza, vaccine uptake in Merseyside was also below the DH target of 55%, with 853/989 (86%) LSOA areas not meeting the target in 2015/2016 (figure 4C). Vaccine uptake was lowest in the most socioeconomically deprived quintiles (figure 5). In the 2014/2015 and 2015/2016 seasons, the odds of influenza vaccine uptake in the most deprived quintile compared with the least deprived were 30% lower in the 24–59 months group (OR 0.70; 95% CI 0.66 to 0.74; p<0.001), 10% lower in the 65+ year age group (OR 0.90; 95% CI 0.88 to 0.92; p<0.001). In

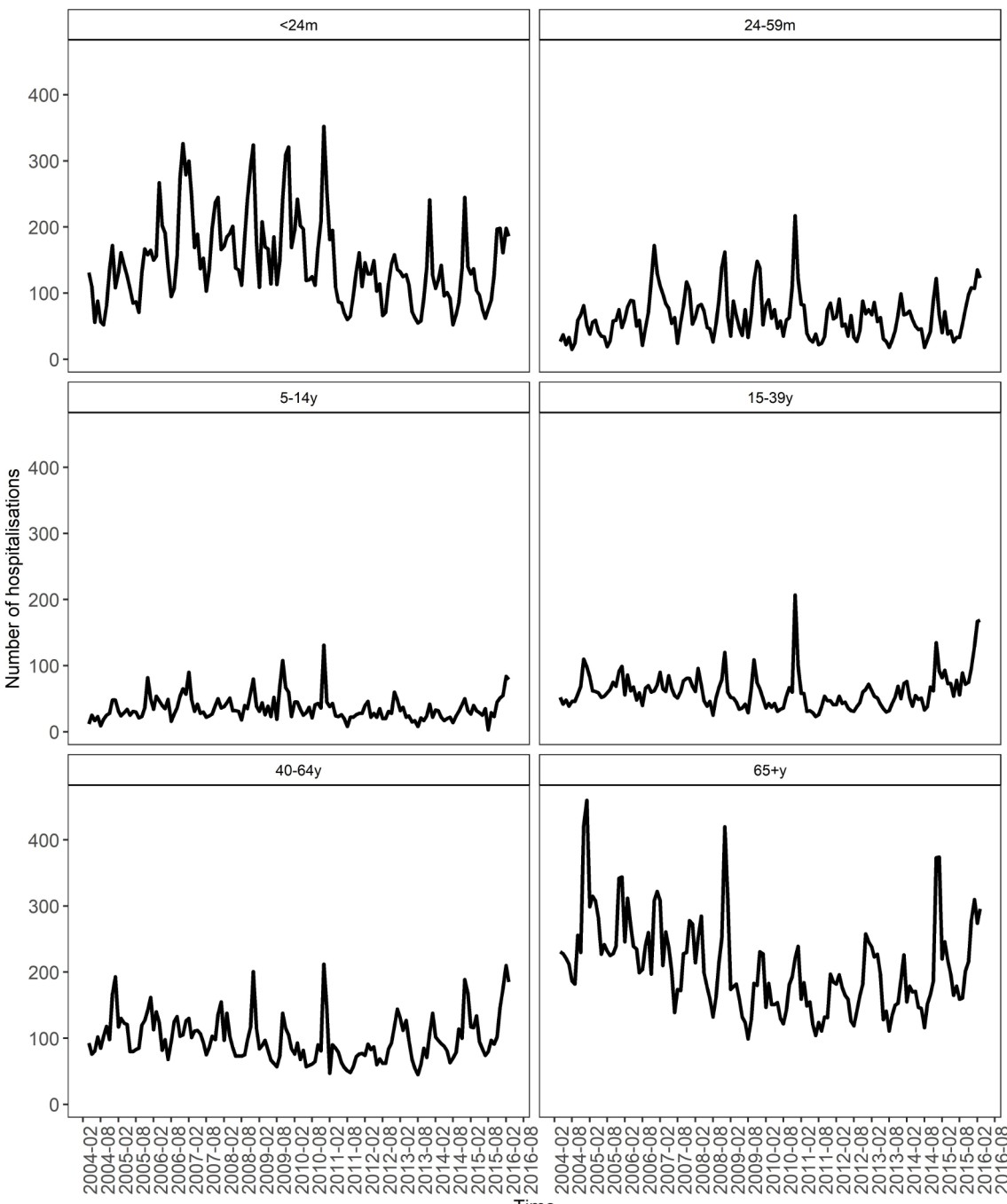

**Figure 1** Number of hospitalisations for influenza-associated illness among Merseyside residents, 2004/2005–2015/2016, by month and age group. m, months; y, years.

at-risk groups, there was no major difference, 3% lower in most deprived (OR 1.00; 95% CI 0.98 to 1.04; p=0.66).

## DISCUSSION

This study shows variations in the number of hospitalisations for IAI in Merseyside, England, across the influenza seasons from 2004/2005 to 2015/2016, with a significant measurable difference by socioeconomic deprivation status in the adult population, which is in concordance with findings from other settings.[8 30 31] Hospitalised IAI rates among those aged 40–64 years living in the most

deprived communities were more than double those of the same age group from the most affluent areas. Influenza vaccine uptake was below the national DH target among those aged <65 years presenting comorbidities, and while it was close to the 75% target for the 65+ years age group in 20014/2015, most of Merseyside was below target in 2015/2016.

In contrast to our findings among adults, IAI hospitalisation rates among children <5 years of age were similar across socioeconomic strata, and vaccination uptake remained consistently below the 40% DH target for both

**Table 1** Population of Merseyside admitted to hospital with IAI by national IMD quintile between 2004/2005 and 2015/2016 (n=89058)

| | National IMD quintile | | | | | | | | | | | |
| | 5 (least deprived) (n=5454) | | 4 (n=10 567) | | 3 (n=12 189) | | 2 (n=13 840) | | 1 (most deprived) (n=47 008) | | Total | |
| | n | % | n | % | n | % | n | % | n | % | n | % |
| **Sex** | | | | | | | | | | | | |
| Female | 2773 | 51 | 5394 | 51 | 6192 | 51 | 7121 | 51 | 23 809 | 51 | 45 289 | 51 |
| Male | 2681 | 49 | 5173 | 49 | 5997 | 49 | 6719 | 49 | 23 197 | 49 | 43 767 | 49 |
| Unknown | 0 | 0 | 0 | 0 | 0 | 0 | 0 | 0 | 2 | 0 | 2 | 0 |
| **Age group** | | | | | | | | | | | | |
| <24 m | 1205 | 22 | 2344 | 22 | 2898 | 23 | 3195 | 23 | 11 779 | 25 | 21 421 | 24 |
| 24–59 m | 607 | 11 | 1115 | 11 | 1290 | 10 | 1444 | 10 | 4917 | 10 | 9373 | 11 |
| 5–14 y | 385 | 7 | 679 | 6 | 634 | 6 | 801 | 6 | 2629 | 6 | 5128 | 6 |
| 15–39 y | 343 | 6 | 769 | 7 | 1079 | 9 | 1290 | 9 | 5399 | 11 | 8880 | 10 |
| 40–64 y | 742 | 14 | 1517 | 14 | 1883 | 15 | 2131 | 15 | 8096 | 17 | 14 369 | 16 |
| 65+ y | 2172 | 40 | 4143 | 39 | 4405 | 36 | 4979 | 36 | 14 188 | 30 | 29 887 | 34 |
| **Comorbidity** | | | | | | | | | | | | |
| No | 2782 | 51 | 5297 | 50 | 6389 | 51 | 7037 | 51 | 24 841 | 53 | 46 346 | 52 |
| Yes | 2672 | 49 | 5270 | 50 | 5800 | 49 | 6803 | 49 | 22 167 | 47 | 42 712 | 48 |
| **Comorbidity by age group*** | | | | | | | | | | | | |
| <24 m | 73 | 6 | 148 | 6 | 187 | 6 | 202 | 6 | 841 | 7 | 1451 | 7 |
| 24–59 m | 112 | 18 | 216 | 19 | 231 | 18 | 302 | 21 | 1126 | 23 | 1987 | 21 |
| 5–14 y | 141 | 37 | 268 | 39 | 231 | 36 | 349 | 44 | 1126 | 43 | 2115 | 41 |
| 15–39 y | 145 | 42 | 361 | 47 | 479 | 44 | 625 | 48 | 2598 | 48 | 4208 | 47 |
| 40–64 y | 439 | 59 | 961 | 63 | 1183 | 63 | 1349 | 63 | 5292 | 65 | 9224 | 64 |
| 65+ y | 1762 | 81 | 3316 | 80 | 3489 | 79 | 3976 | 80 | 11 184 | 79 | 23 727 | 79 |

*Denominator for percentages is the total number of hospitalisations in each age group for each deprivation quintile.
IAI, influenza-associated illness; IMD, Index of Multiple Deprivation; m, months; y, years.

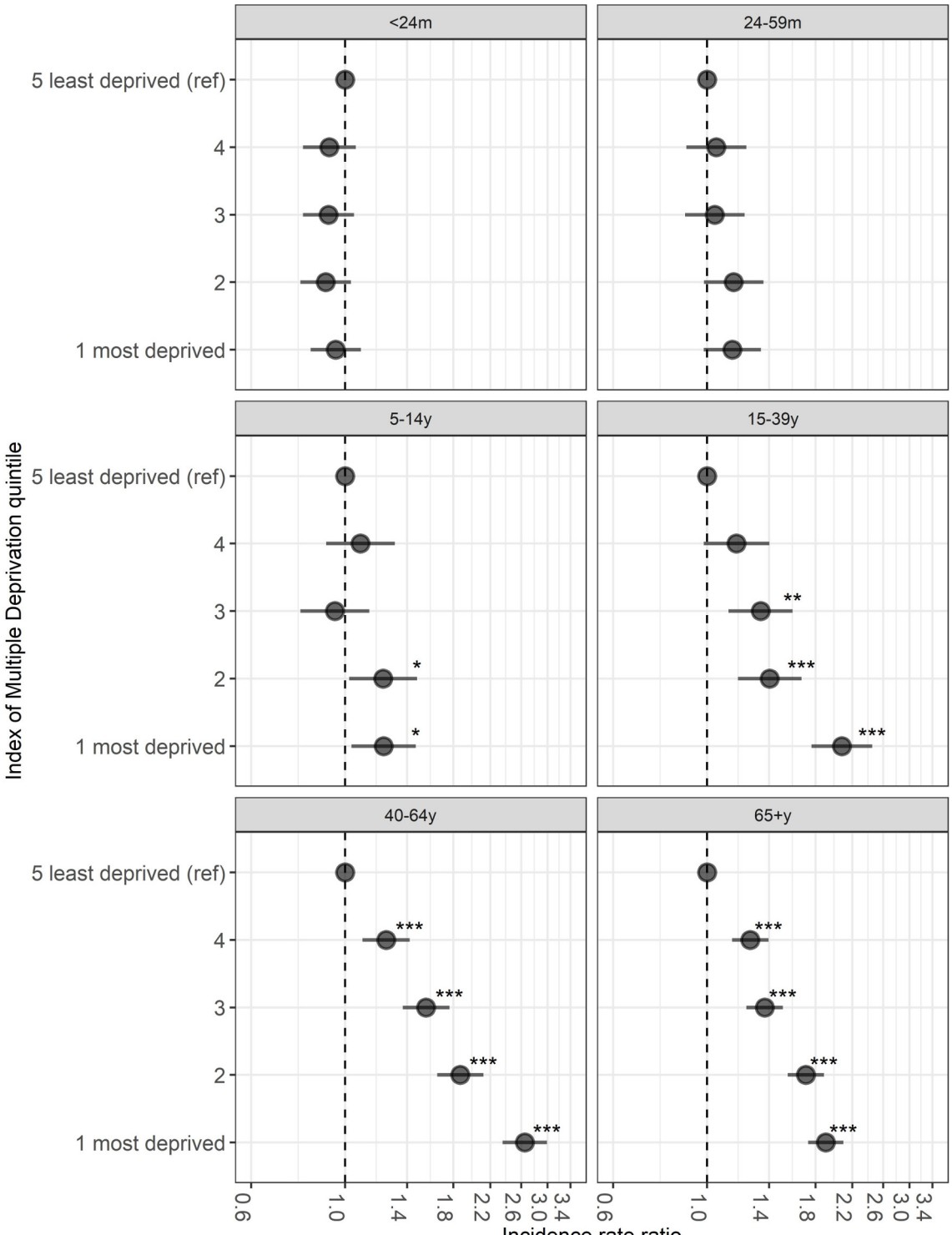

**Figure 2** Incidence rate ratios of influenza-associated illness (IAI) hospitalisation rate for Index of Multiple Deprivation quintiles 1 (most deprived)–4 compared with a reference group of the least deprived quintile (5) by age group. Rate ratios for IAI hospitalised incidence rates were estimated with negative binomial regression stratified by age group. Models were additionally adjusted for sex and fiscal year. m, months; y, years. *p<0.05, **p<0.01, ***p<0.001.

2014/2015 and 2015/2016 seasons. It could be hypothesised that in childhood, in the absence of comorbidities that could exacerbate the disease, hospitalisation following exposure to the influenza virus is likely to be driven by factors other than social determinants of health. While there is robust evidence indicating that social and

economic disadvantage is strongly associated with the development of ill-health in childhood and in later life,[32] our results suggest that hospitalisations for infectious respiratory diseases, such as influenza illness, are an equal threat to very young children regardless of their socioeconomic background, which is likely to be related to there

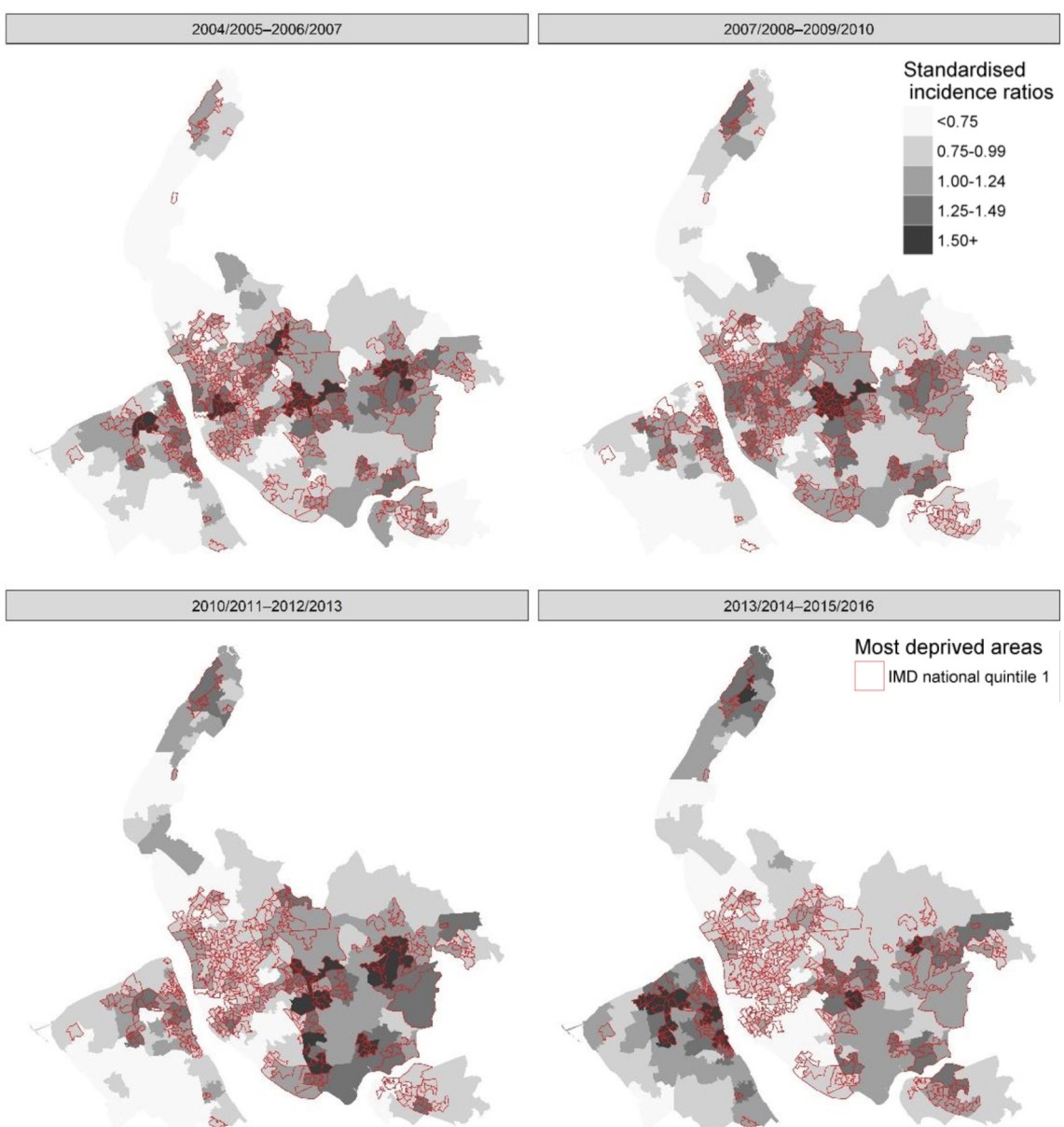

**Figure 3** Age-adjusted and sex-adjusted standardised incidence ratios for influenza-associated illness hospitalisations for Merseyside residents by Middle Super Output Area and pooled years, 2004/2005–2015/2016. IMD, Index of Multiple Deprivation.

being less of a differential in comorbidities across deprivation strata at a young age.

While data on vaccine uptake was not available for influenza seasons prior to 2014/2015, hospitalisation rates have followed a consistent pattern throughout the study period, with an age-shift observed in the 2009/2010 season, consistent with the H1N1 pandemic.[33] Consistent failure to attain the DH vaccination targets across all eligible groups warrant further investigations. Accessibility to vaccination, knowledge and perceptions of both vaccination and disease, are critical aspects for explaining and tackling low uptake of influenza vaccination. The interaction and relative importance of these issues will vary between populations, and will differ between adults and children. There is evidence that individuals at high risk of

complicated influenza infection do not perceive themselves to be susceptible to influenza illness, and that they have concerns over adverse effects of vaccination,[34–36] which suggest that providing knowledge and education about influenza disease to high-risk groups will be crucial for any interventions attempting to improve vaccine uptake. Our findings further highlight the need for informed targeted interventions addressing low vaccine uptake among children and individuals at risk of complicated influenza to ensure national vaccination targets are met in coming years. The lower vaccine uptake rates observed among the most socioeconomically deprived populations compared the most affluent ones suggest that interventions should prioritise areas where vaccine uptake is low and hospitalisation rates are high. Vaccination with the seasonal

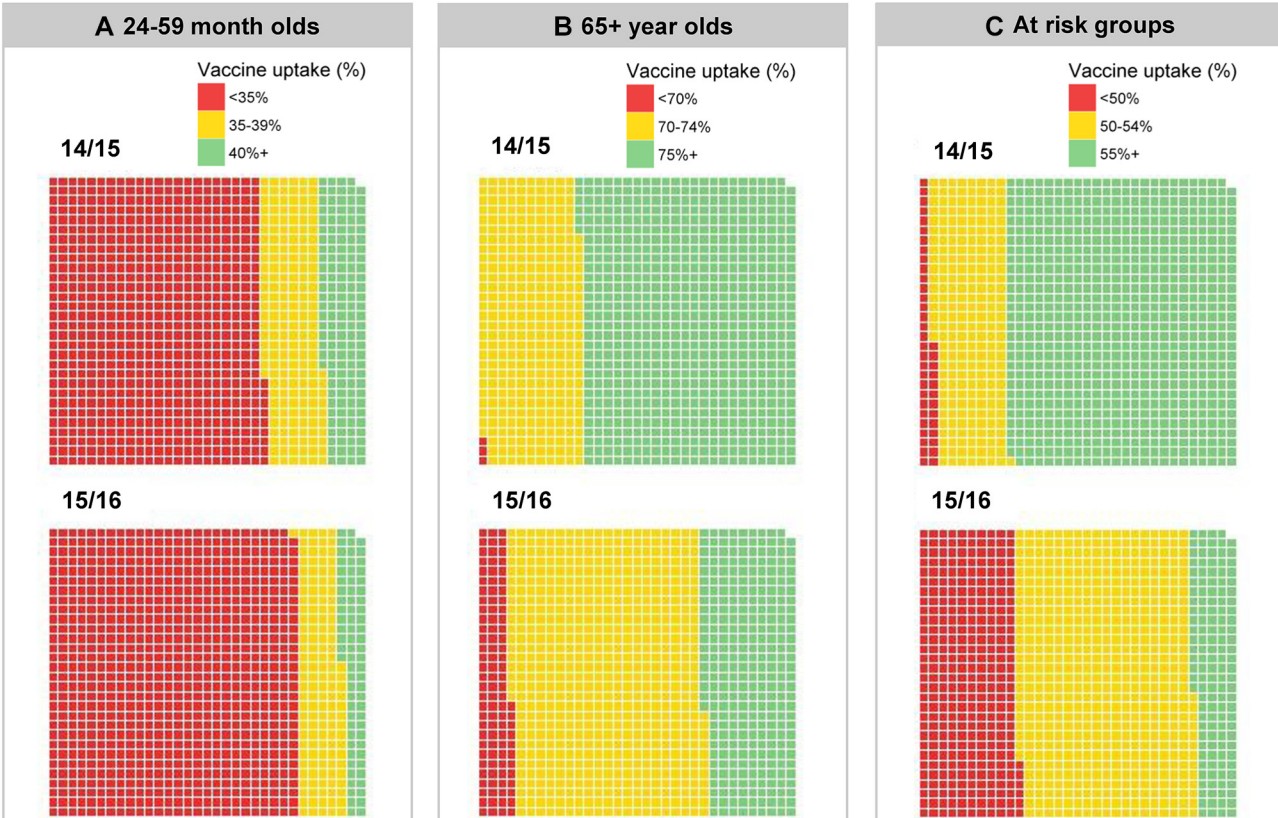

**Figure 4** Influenza vaccine uptake by Lower Super Output Area (LSOA) 2014/2015 and 2015/2016 seasons in those aged (A) 24–59 months (40% NHS England target); (B) 65+ years (75% NHS England target); (C) at-risk groups aged 6 months to 64 years (55% NHS England target). Each block represents 1 of 989 LSOAs in Merseyside.

influenza vaccine can induce long-term cross-reactive antibodies that could have a protective effect in the event of an influenza pandemic.[37 38] Highlighting this added benefit of the vaccine could potentially have a positive influence in people's perception of influenza vaccine, and contribute to improve vaccine uptake.

The strongest association between IAI hospitalisation and deprivation was seen among those aged 40–64 years. This could be in great part explained by the fact that unhealthy behaviours (eg, smoking, alcohol abuse) and environmental risk factors (eg, poor living environment) are associated with lower socioeconomic status. These factors will contribute cumulatively to comorbidities over time peaking in middle to old age,[39] and will, in turn, have a negative impact in on life expectancy, which is lower in deprived areas.[40] While, vaccine uptake in vaccine-eligible patients aged 65+ years is significantly higher than in those at risk aged 40–64 years, the association between IAI hospitalisation and deprivation is still clearly present, although slightly less pronounced. Interventions designed with the aim of improving the overall health and quality of life of the adults in the most socioeconomically deprived populations can potentially contribute to a reduction in IAI.[41]

### Limitations
Reliance on ICD-10 codes for the identification of IAI is an important limitation to our study. While we can assume that ICD-10 coding for confirmed influenza denotes laboratory confirmation, many hospitalised influenza cases will be unconfirmed by laboratory diagnostics and are likely to receive a symptom-based ICD-10 diagnosis code on discharge. Consequently, the true burden of influenza hospitalisations could be higher or lower than our estimations. The use of a seasonal usage ratio to help identify ICD-10 codes associated with IAI was identified a priori through professional consultation but unvalidated against a gold standard diagnostic. However, the selection of ICD-10 codes used for this study was carefully assessed and informed by previous studies,[2 24 25] in an attempt to minimise misclassification. Crucially for the research aims, variations in coding over time were not observed, nor are they likely to be influenced by the socioeconomic background of the patient, since we used data from the national health system. Therefore, we are reassured that the observed association and direction of the effect have not been affected by using ICD-10 coding.

Another important limitation identified in this study is the fact that information is recorded in several databases throughout the health system without common identifiers to enable the data linkage of individuals. While hospitalisation data could be identified to LSOA level through HES, vaccine uptake could only be obtained at GP practice level through ImmForm, and was synthetically estimated at LSOA level. Moreover, none of these

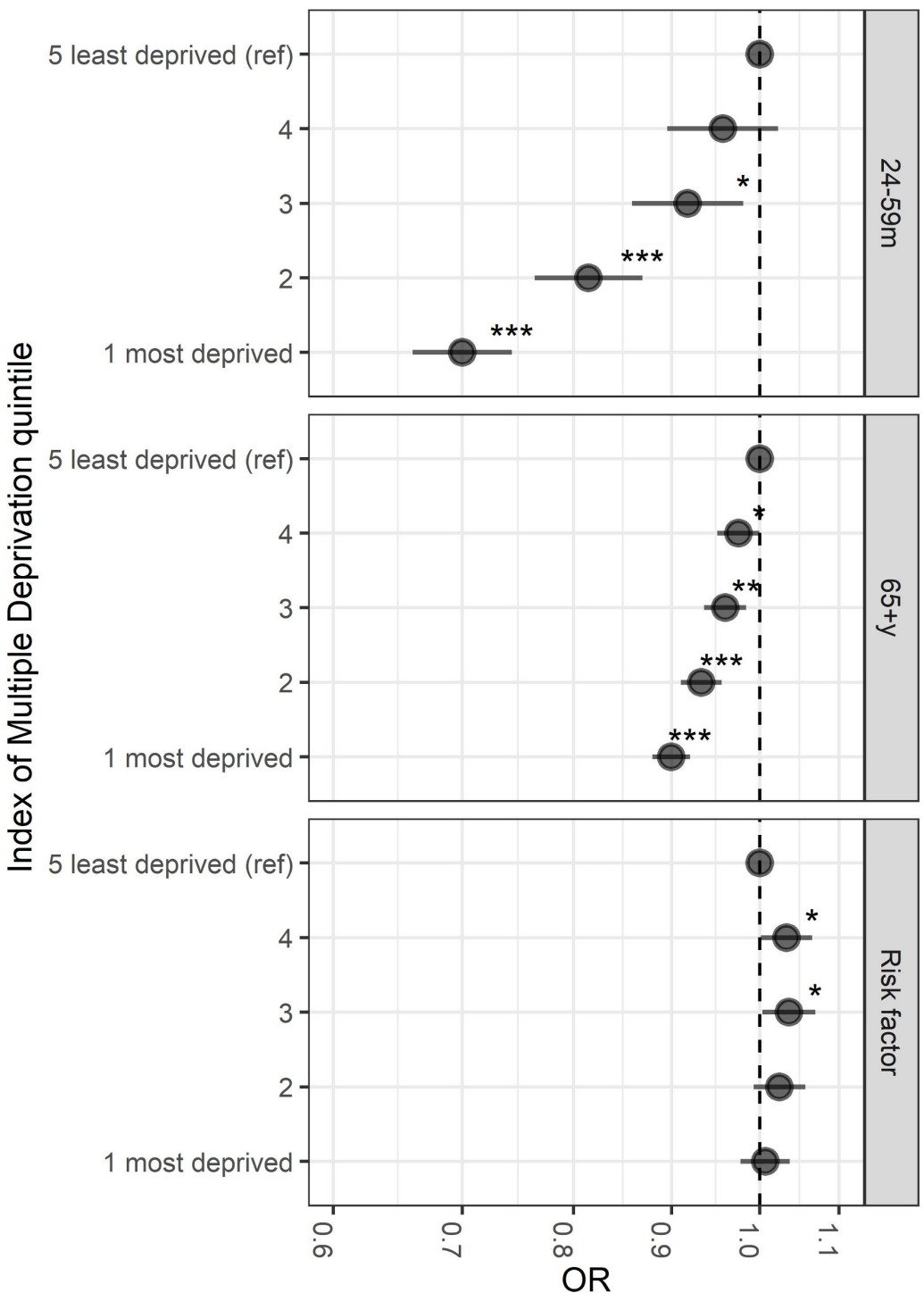

**Figure 5** OR of vaccine uptake by Index of Multiple Deprivation quintiles 1 (most deprived)–4 compared with a reference group of the least deprived quintile (5) by age group, 2014/2015–2015/2016. Adjusted ORs for vaccine uptake were estimated with general linear models of the binomial family with a logit function, robust SEs were used to calculate 95% CIs. Models were additionally adjusted for fiscal year and weighted using the relevant age-related population of each Lower Super Output Area. m, months; y, years. *P<0.05, **p<0.01, ***p<0.001.

databases are linked to laboratory information. While the correlation with socioeconomic deprivation was demonstrated for both outcomes, and is apparent in our graphic representations, this limitation prevented the development of more complex spatiotemporal Bayesian hierarchical models. In order to evaluate interventions aimed at improving vaccine uptake and reducing the burden of hospitalisations due to IAI, surveillance data on disease incidence, hospitalisations and vaccine uptake should be recorded in a systematic and consistent manner over time to facilitate comparison before and after the intervention was implemented. A surveillance system facilitating record matching between the various existing databases and the incorporation of laboratory data would facilitate

recording disease trends and the identification of any changes in disease epidemiology, naturally or as a consequence of a public health intervention.

Finally, while our study has been able to demonstrate the association between socioeconomic deprivation and IAI hospitalisations, and with lower vaccine uptake, the results cannot elucidate the underlying causes for this observation. Comorbidities exacerbate respiratory disease,[6] and socially deprived areas tend to have more limited access to social resources, resulting in poorer health outcomes.[42] Therefore, designing effective vaccine policy and public health interventions that translate outcomes to health equity is critical; and requires improved understanding of the healthcare system engagement that underpins lower influenza vaccine uptake.[43]

**Acknowledgements** The authors acknowledge the contribution of Gabbie Marr and Ed Gaynor at Liverpool NHS Clinical Commissioning Group and Emer Coffey from Public Health, Liverpool City Council. This work uses data provided by patients and collected by the NHS as part of their care and support and would not have been possible without access to this data. The NIHR recognises and values the role of patient data, securely accessed and stored, both in underpinning and leading to improvements in research and care.

**Contributors** AI-P, DH, PC and NF conceived of and designed the study. DH acquired and analysed the data. All authors interpreted the data; DH and AI-P wrote the first draft of the report and all authors reviewed the draft and final manuscript.

**Funding** This study was supported by a grant from NIHR Research Capacity Funding via Liverpool NHS clinical commissioning group. The views expressed are those of the authors and not necessarily those of the NHS, the NIHR, the Department of Health or Public Health England.

**Competing interests** NF and DH were in receipt of research grant support on the topic of rotavirus vaccines from GlaxoSmithKline (GSK) Biologicals. DH is in receipt of research grant support on the topic of rotavirus vaccines from Sanofi Pasteur, and Merck & Co (Kenilworth, New Jersey, USA) after the closure of Sanofi Pasteur-MSD in December 2016.

**Patient consent** Not required.

**Ethics approval** The data accessed were aggregated and anonymised and therefore as specified by the University of Liverpool ethics guidance and the NHS Health Research Authority Research Do I need NHS REC approval tool (http://www.hra-decisiontools.org.uk/research/), this study did not require ethical approval.

**Provenance and peer review** Not commissioned; externally peer reviewed.

**Data sharing statement** The data that support the findings of this study are available from a Public Health England but restrictions apply to the availability of these data as they are not publicly available. Aggregated data may be available from the authors/Public Health England on reasonable request and with permission of Public Health England.

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
