## [Reviewer comments · BMJ Open]

ARTICLE DETAILS

TITLE (PROVISIONAL)	Influenza associated hospitalisation, vaccine uptake and socioeconomic deprivation in an English city region: an ecological study
AUTHORS	Hungerford, Daniel; Ibarz-Pavon, Ana; Cleary, Paul; French, Neil

VERSION 1 – REVIEW

REVIEWER	Amalie Dyda Australian Institute of Health Innovation, Macquarie University, Sydney, Australia
REVIEW RETURNED	27-Jun-2018

GENERAL COMMENTS	This is an excellent paper, very well written and contributes to the literature in the area of factors that impact the burden of influenza. The paper describes an ecological study investigating the association between socioeconomic deprivation and hospitalisation due to influenza or an associated illness. In particular, the statistical methods are very thoroughly considered and described. There are some limitations to this study design, particularly in relation to the use of ICD10 codes and vaccination data from GPs, but these have been appropriately considered in the discussion. Some minor comments for consideration: Ethics statement: The authors state that ethical approval was not required for this study as aggregate and anonymised data were used. I would suggest this should still be checked with an ethics committee, and if they agree ethics is not required then this should be stated. Introduction: Line 47- Are these flu data based on surveillance data? The authors go on to explain that this is likely an underestimate but some clarification would help. Line 56- are the vaccines funded? i.e. provided free of charge. Methods: Line 95- It might be worth mentioning where people get vaccinated. Are there places other than GPS that people are likely to get vaccinated that are not captured in these data? Discussion: Line 270- 'warrant further investigations'. I think this sentence needs some clarification. What has already been done? Not sure if further research in to attitudes and beliefs is what is needed. In addition, the issues relating to childhood and adult vaccination are very different.
--

	Knowledge and access more likely to affect adult populations. Line 280- Strongest association in 40-64 years. It is hypothesised that this is due to increasing age which I agree with. But if this is the case, why is 65+ not the strongest association? It would be good to mention higher vacc uptake in this age group and also that you did find a similar trend in this age group just not as strong.
REVIEWER	JEFFERY CUTTER MINISTRY OF HEALTH, SINGAPORE
REVIEW RETURNED	29-Jun-2018
GENERAL COMMENTS	This is a well designed and conducted study. The manuscript was very well written. It was a pleasure to read. Well done.

VERSION 1 – AUTHOR RESPONSE

Reviewer #1

Thank you for your comments, we believe the associated edits have helped to enhance the manuscript.

Ethics statement:

Full clarification of the studies exemption from ethics has been added to the ethics statement at the end of the manuscript.

Line 47- Are these flu data based on surveillance data? The authors go on to explain that this is likely an underestimate but some clarification would help.

We have expanded on the data sources used in the cited study.

Line 56- are the vaccines funded? i.e. provided free of charge.

The vaccine is provided free of charge in eligible groups through the National Health Service, this has been added to the background section.

Line 95- It might be worth mentioning where people get vaccinated. Are there places other than GPs that people are likely to get vaccinated that are not captured in these data?

This is an important point, vaccines can be given in pharmacies and these vaccinations should be captured through the GP system. We have therefore added a reference and clarification to the methods section.

Line 270- ‘warrant further investigations’. I think this sentence needs some clarification. What has already been done? Not sure if further research in to attitudes and beliefs is what is needed. In addition, the issues relating to childhood and adult vaccination are very different. Knowledge and access more likely to affect adult populations.

We agree, so we have added further clarification, including references from studies looking at knowledge and attitudes towards influenza disease and influenza vaccination.

Line 280- Strongest association in 40-64 years. It is hypothesised that this is due to increasing age which I agree with. But if this is the case, why is 65+ not the strongest association? It would be good to mention higher vacc uptake in this age group and also that you did find a similar trend in this age group just not as strong.

We have now added the association in 65+ year olds to the discussion, the strength of the association is likely to be related to vaccine uptake and the likelihood that an age above 65+ years, socioeconomic deprivation's association with poor health outcomes becomes reduced in comparison to the effect of getting older and more vulnerable.

Reviewer #2

Thank you for reviewing the manuscript and your kind comments.

VERSION 2 – REVIEW

REVIEWER	Amalie Dyda Australian Institute of Health Innovation, Macquarie University, Australia
REVIEW RETURNED	27-Aug-2018
GENERAL COMMENTS	As mentioned this is a very well researched and well written paper. The authors have adequately addressed all previous comments and I would recommend this paper for publication without further amendments.